# Influence of 1-Methylcyclopropene (1-MCP) on the Processing and Microbial Communities of Spanish-Style and Directly Brined Green Table Olive Fermentations

Elio López-García, Antonio Benítez-Cabello, Francisco Rodríguez-Gómez, Virginia Martín-Arranz, Antonio Garrido-Fernández and Francisco Noé Arroyo-López *

Food Biotechnology Department, Instituto de la Grasa(CSIC), Universidad Pablo de Olavide, 41013 Seville, Spain
* Correspondence: fnoe@ig.csic.es

**Abstract:** This work evaluates the effect of 1-methylcyclopropene (1-MCP) on postharvest and fermentation of Manzanilla cultivar, processed as Spanish-style or directly brined table olives. During postharvest handling, 1-MCP (2.85 μL/L) reduced the number of colour-turning olives by 18.42% over the untreated fruits. In Spanish-style and directly brined fermentation, the 1-MCP treatment led to lower pH levels, higher titratable acidities, improved firmness and colour olives than untreated fruits. A panel of expert testers also gave higher scores, and overall acceptability to the 1-MCP treated fruits, especially in the case of Spanish-style fermented olives. Metagenomic analysis of olive biofilms at the end of the fermentation process (176 days) revealed that *Lactiplantibacillus* was the most abundant bacterial genus in both Spanish-style and directly brined olives (>72%). However, fungal biodiversity was higher than bacterial in all treatments. *Saccharomyces* was the predominant yeast genus associated with directly brined olives (>97%), whilst *Wickerhamomyces* (>37%) and *Zygoascus* (>18%) were with Spanish-style fermentations. The 1-MCP treatment doubled the presence of *Wickerhamomyces* in Spanish-style fruits (74%) whilst reducing the presence of *Zygoascus* and allowing the growth of *Enterobacter* (15%) in directly brined olives. Thus, the postharvesting treatment of table olives with 1-MCP could help reduce the maturation progress of olives and improve the organoleptic and quality characteristics of the products without affecting the microbiological evolution of the fermentations.

**Keywords:** table olives; fermentation; 1-MCP; metagenomic; Spanish-style; natural olives

## 1. Introduction

The olive tree, *Olea europaea* L., is a member of the *Oleaceae* family, native to the eastern part of the Mediterranean region. Depending on the olive variety, the fruits can be used for oil extraction, table olive, or both (double use). Table olives are one of the most important fermented vegetables in the Mediterranean countries, with an annual world production of approximately $3 \times 10^6$ tons/year [1]. Olive fruits must be processed to eliminate their natural bitterness, occasioned by the presence of the bitter glucoside oleuropein [2]. Alkali-treated olives (Spanish style), ripe olives by alkaline oxidation (Californian style), and directly brined olives (natural black or green olives) are the most common processing methods [3].

It has been proven that the reduction of ethylene production delays ripening and senescence in several species of climacteric fruits [4,5]. Olive fruits did not show a climacteric respiratory behaviour and did not exhibit a softening or anthocyanin synthesis after harvest in response to ethylene treatment [6]. Crisosto et al. [7] reported that green olives produced very little ethylene but could be moderately sensitive to ethylene action. By contrast, mature black olives released significantly higher quantities but still very low compared to climacteric fruits. A concentration of 150–250 μL/L ethylene only increased the respiration rate of green olives at 20 °C slightly but considerably increased respiration rates at 25 or 30 °C with a climacteric-type rise depending on the olive cultivar [8].

Conversely, Kafkaletou et al. [9] reported that adding 1000 μL/L ethylene reduced the respiration of green Conservolea fresh olives and promoted an increase in firmness at 20 °C during the first two days after harvesting. In addition, ethylene concentrations up to 1000 μL/L, applied at 25 °C for 10 days to green harvested olives, resulted in firmness retention and a minor decrease in green colour [10]. Non-climacteric fruits do not show dramatic respiration or ethylene production. Moreover, usually, they do not continue maturation after harvest but, instead, undergo senescence parallel to some of the processes occurring in ripening fruit [11].

Slowing the process of ripening and senescence extends the storage, shelf life, and quality of fresh fruit and vegetables. The skin colour and flesh firmness of the olive fruit at the time of processing will determine the quality of the final product. However, both attributes greatly decay during postharvest handling of table olives. Therefore, the industry continues to search for new treatments to improve the fermented olives' texture and colour. For example, 1-methylcyclopropene (1-MCP) has been used to extend the storage life and quality of plant tissue as an inhibitor of ethylene production [12]. Control atmosphere associated with 1-MCP or dynamic controlled atmosphere with low oxygen partial pressures drastically reduced the metabolism of apples, allowing the storage at high temperatures [13]. Applying, at low temperature, 1–MCP treatment to postharvest French prune (*Prunus domestica,* L.) preserved fruit hardness (20%), titratable acidity (29%), ascorbic acid (18%), total soluble solids (21%), and anthocyanins.

By contrast, Xiong et al. [14] retarded moisture loss (44%), colour change, and harvest ripening (7 days). Previous experiments with different olive cultivars showed that 1-MCP treatment effectively reduced colour changes and delayed firmness losses during 15 weeks of storage. The success of the 1-MCP treatment depended on the methods of application, duration, and concentration, as well as commodity factors such as olive variety [15,16]. Kafkaletou and Tsantili [9] also showed that using 1-MCP prevented the loss of green colour in harvested dark green Conservolea olives. However, the effect on the other fruits' processing as table olives and, mainly, on the fermentation processes have not yet been evaluated.

In summary, olives generally behave peculiarly in response to ethylene and 1-MCP. Therefore, further investigations are necessary to elucidate such properties. This work assesses the effects of 1-MCP on postharvested Manzanilla olives and its influence on their further processing as Spanish-style or directly brined (natural) table olives. For this purpose, the fruits' changes during postharvest handling, the physicochemical and microbiological parameters of the fermentation, as well as the sensory and metagenomic characteristics of the fermented products were monitored.

## 2. Materials and Methods

### 2.1. Experimental Conditions

Eighty kg of Manzanilla fruits were hand-harvested in Huevar del Aljarafe (Seville, Spain) at the green ripening stage during October 2020 and transported at 25 °C to the laboratory of Instituto de la Grasa (CSIC, Seville, Spain) in less than 1 h. One part of them (40 kg) was used as control (C) whilst the rest (40 kg) were subjected to a 2.85 μL/L 1-MCP treatment in a closed container (220 L volume) at 25 °C for 20 h, following the manufacturer's recommendations (AgroFreshIbérica, Lleida, Spain). Then, 1 kg of untreated (control) and 1-MCP treated fruits (1-MCP) were kept at room temperature (25 °C) for the determination of turning colour and damaged fruits (*n* = 200 olives) percentages after 24 and 168 h. This task was carried out by 4 members of Instituto de la Grasa staff, all experts in evaluating table olive quality.

The rest of the fruits (39 kg of 1-MCP treated and 39 kg of untreated olives) were processed as Spanish-style (SS) or directly brined (DB) olives. The SS fruits were treated with a 2% NaOH solution for 4.5 h, reaching the lye 2/3 pulp thickness. Then, the fruits were washed (3 h) to remove excess alkali and brined in an 11% (*w/v*) NaCl solution with 0.37% (*v/v*) HCl. In the case of DB, after washing the fruits with tap water to remove

impurities, they were directly brined in a non-acidic 5% NaCl solution. Fermentations were carried out in containers of 8 L volume with 4.5 kg of olives and 3.0 L of brine. All fermentations were inoculated within the first week of the process with the commercial inocula OleicaStarter Advance (TAFIQS in Foods, Seville, Spain), a mix of three strains of *Lactiplantibacillus pentosus* species, and OleicaStarter Yeast (TAFIQS in Foods), itself a blend of the yeast *Wickerhamomyces anomalus* and *Saccharomyces cerevisiae*. Thus, the experimental design consisted of a total of 4 different treatments SS-C (control), Spanish-style fruit; SS-1MCP, Spanish-style 1-MCP treated fruits; DB-C (control) directly brined fruits; and DB-1MPC, directly brined 1-MCP treated fruits. Each treatment was run duplicated (*n* = 8).

### 2.2. Physicochemical Monitoring

Olive brines from the 8 fermentation vessels were sampled at 6, 12, 16, 19, 22, 34, 47, 68, 121, and 176 days of fermentation for determination of pH, NaCl (%), titratable acidity (expressed as g of lactic/100 mL of brine) and combined acidity (expressed as mEq of HCL acid added to 1 L of brine to reach pH 2.6), using an automatic titrator model Excellence (Mettler Toledo, Columbus, OH, USA) and the methods described by Garrido-Fernández et al. [2]. Olive samples from each fermentation vessel were randomly taken to analyse the firmness, surface colour, and moisture of fruit parameters at 0, 16, 34, 68, 121, and 176 days following the methods described elsewhere [17,18]. Colour was measured using a spectrophotometer Model CM-5 (Konica Minolta Sensing Americas, Ramsey, NJ, USA). Interference by stray light was minimised by covering the samples with a box with a matt black interior. Colour was expressed as the CIE LAB parameters for the calculus of hue angle. The firmness of the olives was measured using a Kramer shear compression cell coupled to a Food Texture Analyzer FTM-50 (Techlab Systems, Lezo, Spain). The crosshead speed was 200 mm/min. The firmness, expressed as kN/100 g flesh, was the mean of 10 replicate measurements performed on 3 pitted olives. Moisture content was determined in duplicate by drying 20 g of crushed olive flesh in an oven model Selecta DigiHeat (J.P. Selecta, Barcelona, Spain) at 102 °C until weight stabilisation. Individual reducing sugars (glucose, fructose, sucrose, and mannitol) were determined in an HPLC system at the end of fermentation (176 days), according to the methods developed by Sánchez et al. [19]. The system was composed of a pump model Jasco PU-2089, an autosampler module model AS-2055 (Jasco, Tokyo, Japan), a detector model Varian ProStar 350 RI, a thermostatted Column Compartment TCC-100, which includes column and heater (Dionex, Waltham, MA, USA), a hardware interface between the PC and the components system model LC Net II/ADC (Jasco, Japan), and the software ChromNav (Jasco, Japan) for analysing the data.

### 2.3. Microbial Monitoring

Olive brines were sampled at 0, 6, 12, 16, 22, 34, 47, 68, and 176 days of fermentation for the counts of the *Enterobacteriaceae*, yeasts, and lactic acid bacteria (LAB) populations. Samples drawn from the different treatments assayed were spread onto selective media according to the methods described by Rodríguez-Gómez et al. [20], using a spiral plate maker model easySpiral Dilute (Interscience, Saint Nom la Brétèche, France). Counts were determined using an automatic image analysis system model Scan4000 (Interscience, Saint Nom la Brétèche, France) and expressed as $\log_{10}$ CFU/mL. Microorganisms adhered to the olive epidermis were also determined at the end of the fermentation period (176 days). For this purpose, fruits were removed from the fermentation vessels under sterile conditions and washed twice in sterile distilled water to remove non-adhered cells. Then, fruits were pitted, and 25 g was immediately transferred into a stomacher bag containing 75 mL of a sterile saline solution (0.9% NaCl). The flesh was homogenised for 2 min at maximum speed (300 rpm) in a stomacher model Seward 400 (Seward Medical, Ltd., West Sussex, UK). Suspension of the appropriate dilutions was then spread onto selective media to determine LAB, yeasts, and *Enterobacteriaceae* populations. Counts were expressed as $\log_{10}$ CFU/g.

### 2.4. Sensory Evaluation

After 176 days of fermentation, the fruits obtained from the different treatments were washed (12 h) in tap water and then packaged in polyethylene terephthalate (PET) vessels (1.6 L volume). The packages were filled with 0.9 kg of olives and 0.7 L of new cover brine to obtain in the equilibrium a pH of 3.8, a concentration of 4.0% NaCl, and 0.5% titratable acidity, to preserve the product according to their physicochemical characteristics.

The evaluation sheet developed by the International Olive Council [21] for scoring acidic, saltiness, bitterness, and hardness attributes were used in the present study. Moreover, other attributes such as browning, appreciation of defects, and overall acceptability were also introduced into the evaluation sheet. The panel was composed of 4 expert members from the Instituto de la Grasa (CSIC) staff, chosen because of their usual involvement in previous sensory analyses. The evaluation sheet consisted of two sections. The first was devoted to the sample and panellist identification, while the second included the attributes to be evaluated, including a final question on overall acceptability. During preselected sampling periods, the olives were offered to panellists using blue glass according to the recommendations of the standard COI/T.20/Doc.No 5 (Glass for oil tasting) [22], coded with three digits randomly chosen. All the attributes were evaluated on an unstructured scale which ranged from 1 to 11, in which 1 was associated with the complete absence of the attribute and 11 with its presence in the highest intensity. The panellists were asked to mark on the scale according to the intensity perceived of each attribute. The panel leader read the sheets with 0.1 cm precision.

### 2.5. Metagenomic Analysis

Olive samples from each fermentation vessel were taken at the end of fermentation (176 days), washed in sterile water, and pitted in sterile conditions. Then, 25 g pitted olives was homogenised in 100 mL of sterile saline solution (0.9% NaCl) in a Stomacher® homogenizer (Seward Medical, Ltd., West Sussex, UK) for 5 min and spun at $9000 \times g$ for 15 min. The supernatant was withdrawn, and the pellets were washed twice with sterile saline solution before storing at $-80\,^{\circ}\text{C}$ until DNA extraction. The total genomic DNA from olives was extracted and purified using the PowerFood Microbial DNA Isolation Kit (MoBio, Carlsbad, CA, USA) according to the manufacturer's instructions and sent for sequencing to FISABIO (Valencia, Spain). Before sequencing, purified DNA content was measured using a Qubit fluorometer (Thermo Fisher Scientific, Waltham, MA, USA), always obtaining values above 0.2 ng/µL. Each sample ($n = 8$) was sequenced to determine the structure of the bacterial and fungal populations.

For bacteria, the V3 and V4 regions (459 bp) of the 16S ribosomal RNA gene were amplified with the designed primers surrounding conserved regions [23], following the procedure described by the Illumina amplicon libraries protocol. The DNA amplicon libraries were generated using a limited PCR cycle: initial denaturation at 95 °C for 3 min, followed by 25 cycles of annealing (95 °C for 30 s, 55 °C for 30 s, 72 °C for 30 s), and a final extension at 72 °C for 5 min, using a KAPA HiFi HotStartReadyMix (KK2602) (Roche, Basel, Swiss). For the fungal populations, DNA samples were submitted to PCR-amplification of the ITS1 region located inside the fungal nuclear ribosomal DNA (rDNA) with the designed primers surrounding conserved regions ITS1-F_KYO2 (18S SSU 1733–1753) and ITS2_KYO2 (5.8 2046–2029) [24]. The DNA amplicon libraries were generated using the following limited PCR cycle: initial denaturation at 95 °C for 3 min, followed by 28 cycles of annealing (95 °C for 30 s, 55 °C for 30 s, 72 °C for 30 s), and a final extension at 72 °C for 5 min, using a KAPA HiFi HotStartReadyMix(KK2602) (Roche, Basel, Swiss). Then, in both cases, the Illumina sequencing adaptors and dual-index barcodes (Nextera XT index kit v2, FC-131-2001) were added to the amplicons. Libraries were normalised and pooled before sequencing. The pool containing indexed amplicons was loaded on the MiSeq reagent cartridge v3 (MS-102-3003) (Illumine, San Diego, CA, USA) spiked with 25% PhiX control to improve base calling during sequencing, as recommended by Illumina for amplicon

sequencing. Sequencing was conducted using a paired-end, $2 \times 300$ bp cycle run on an Illumina MiSeq sequencing system (Illumine, San Diego, CA, USA).

Data obtained from the 16 sequenced sets of data ($n = 8$ for bacteria and $n = 8$ for fungi) were analysed using NG-Tax [25] under default parameters. For each sample, only the most abundant sequences (>0.01%) were retained as Amplicon Sequence Variant (ASV); the remaining reads were clustered against those ASVs allowing one mismatch to correct for error sequencing. Taxonomy was assigned using the SILVA 138 SSURef database for the 16S rRNA amplicon samples (bacteria) and the full UNITE+INSDC for the ITS amplicon samples (fungi). Plots were generated using ggplot 2 3.3.2 [26] and Metacoder 0.3.4 software packages.

### 2.6. Statistical Analysis

The data were subjected to an analysis of variance (ANOVA). For this purpose, the factorial ANOVA module of Statistica 7.1 software (Statsoft Inc., Tulsa, OK, USA) was used to check for significant differences among physicochemical, microbiological and sensory attributes as a function of the different treatment assayed (SS-C; SS-1MCP; DB-C; DB-1MPC) and time of fermentation (6, 12, 16, 19, 22, 34, 47, 68, 121, and 176 days). A post hoc statistical LSD test was applied using $p \leq 0.05$ as the cut-off level of significance. In addition, the multiple sets of data were also analysed using Multiple Factor Analysis (MFA) (XLSTAT, Addinsoft INC, New York, NY, USA), which is a technique devoted to data tables in which a set of individuals (treatments) is described by several groups of variables [27]. The tool was applied using the package FactoMineR [28] in R software v.4.1.3. Only those variables with correlation to dimensions higher than 0.8 were selected. The data were balanced to prevent the dominance of those variables with high absolute values, using the same weight within each group to preserve the group structure.

### 3. Results and Discussion

In industry, and depending on the processing conditions of each factory and olive variety, harvested fruits can be stored at ambient temperatures for up to 1–2 days, awaiting processing [2]. During this period, respiration and ripening of fruits can continue, leading to a loss of quality, with the effect being particularly accelerated with temperature, losing the green colour at 10 °C and developing purple tones at 20 °C [29]. Studies on Gordal respiration showed a rapid decline during the first two days but continued at a slow rate for several days, with mannitol being the sugar consumed preferably; moreover, loss of weight, colour changes, and respiration activity increase with temperature and olive maturation [30]. Storage of harvested olives in water reduced the respiration activity but increased the intercellular volume because of the $CO_2$ accumulation in the flesh [31].

This work evaluates the influence of 1-MCP treatment on the postharvest handling and processing of Manzanilla fruits as Spanish-style (NaOH treated) and directly brined table olives, comparing the results with the usual processes using untreated fruits (controls). The 1-MCP was applied within the first 6 h after fruit picking, using a concentration of 2.85 µL/L in the surrounding atmosphere. This concentration was chosen according to the manufacturer's recommendations, based on their experience with other similar fruits.

### 3.1. Postharvest Handling and Physicochemical Changes during Fermentation

After treatment, the effect of 1-MCP on the olive fruits was evaluated. Figure 1 shows the evolution of the percentage of colour-turning olives, damaged fruits, and fruit appearance at 24 and 168 h after harvesting. As observed, the application of the 1-MCP reduced the ripening of Manzanilla olives during postharvest storage since the untreated lot had an 18% higher number of colour-turning fruits after 168 h reception (Figure 1A,C). Nevertheless, it did not modify the percentage of damages produced while knocking the olives down to facilitate picking. Amini & Ramin [15] reported that applying 1-MCP on green olives just after reception reduced the rate of ethylene production and respiration, colour changes and fruit softening. Similar results were also obtained by Ramin [16] for the

Conservolea variety, who showed that application of 1-MCP delayed olive softening and colour changes compared to untreated fruits (control) during postharvest storage.

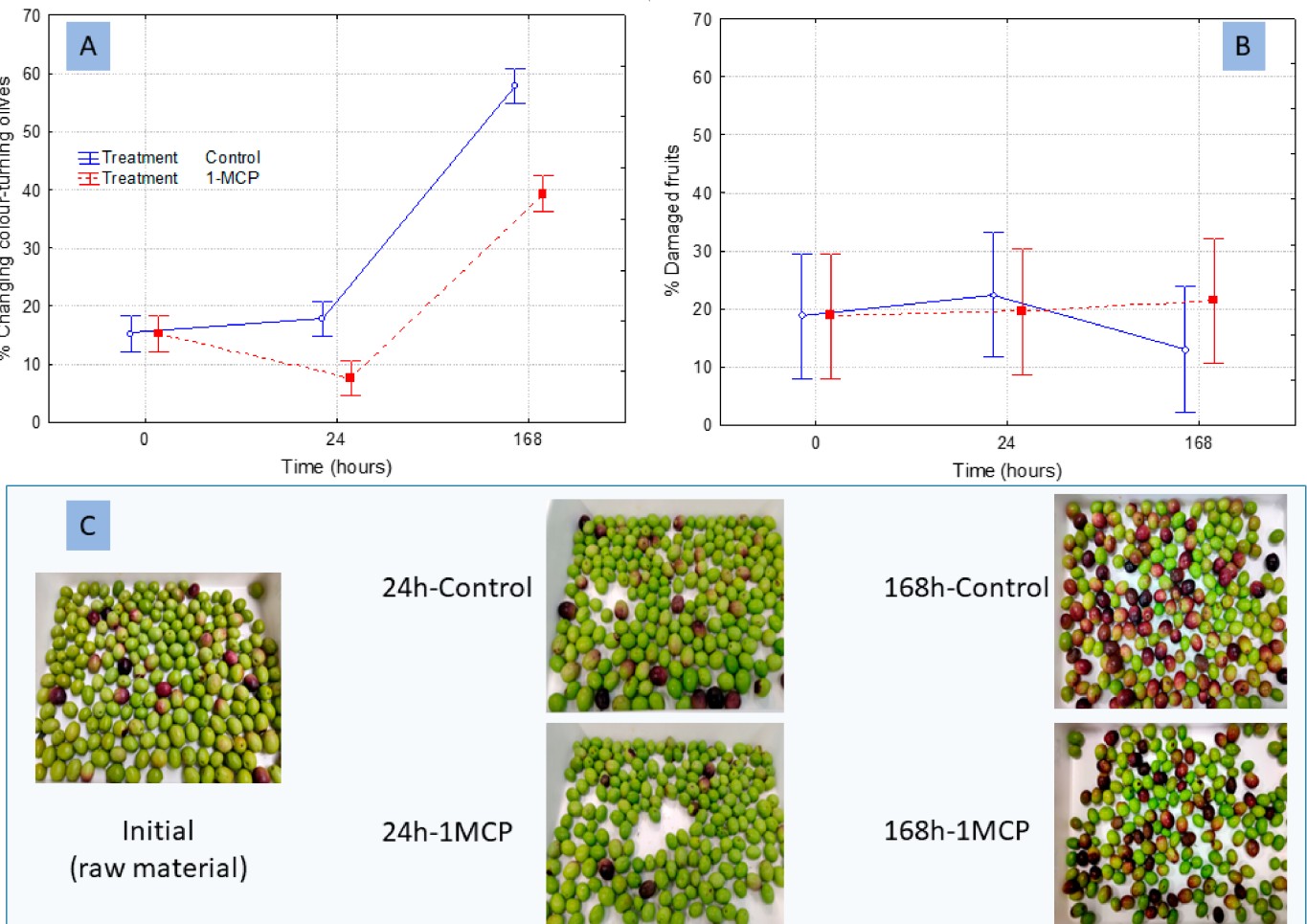

**Figure 1.** Evolution of the percentage of colour-turning olives (**A**) and damaged fruits (**B**), determined by a panel of experts (*n* = 4), as well as fruit appearance (**C**) for the control and 1-MCP treated fruits during the first 168 h after harvesting. Time 0 corresponds to the moment just after picking and before 1-MCP treatment. Non-overlapping error bars stand for significant differences among treatments according to ANOVA analysis and post hoc statistical LSD test using $p \leq 0.05$ as the cut-off level of significance.

Figure 2 shows the evolution of the main physicochemical parameters usually considered when assessing the fermentation course. A clear difference between fruits processed as SS or DB was noticed since they followed the typical trends expected for both types of fermentation [2]. However, statistical differences in pH and titratable acidity between 1-MCP treated and untreated fruits were detected at certain sampling times (Figure 2A,B). Thus, at the fermentation process endpoints, a lower pH and higher titratable acidity levels for SS-1MCP and DB-1MCP treatments compared with their respective controls were noticed (pH SS-C 4.45 vs. 4.00 SS-1MCP; pH DB-C 4.60 vs. 4.50 DB-1MCP; titratable acidity SS-C 0.40% vs. 0.62% SS-1MCP; titratable acidity DB-C 0.36% vs. 0.47% DB-1MCP). During table olive fermentation, LAB produces lactic acid, which causes a titratable acidity increase and consequently a drop in pH [2].

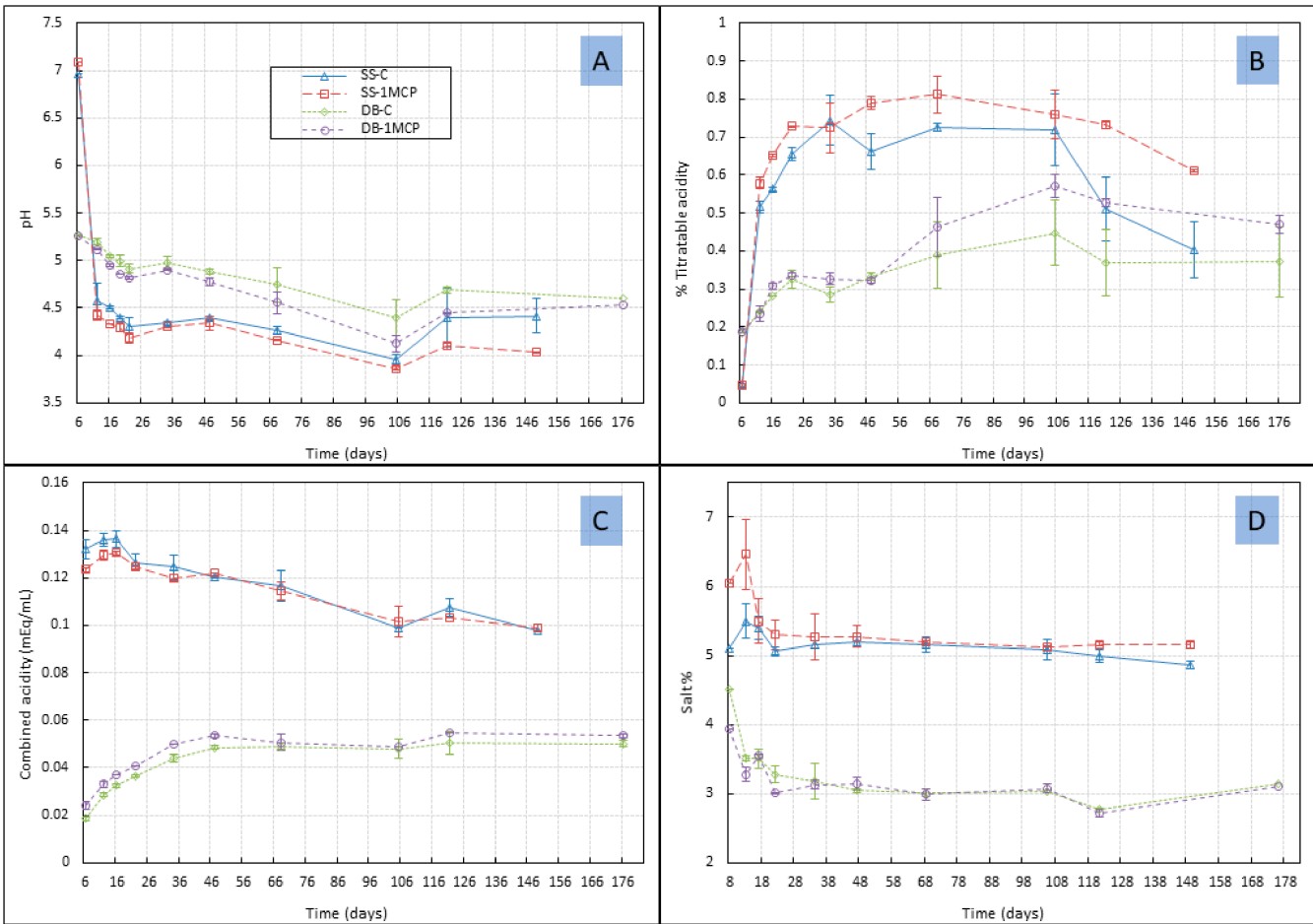

**Figure 2.** Evolution in the brine of pH (**A**), titratable acidity (**B**), combined acidity (**C**), and salt content (**D**) during fermentation in the different treatments assayed in this work. DB-C: control directly brined olives, DB-1MCP: directly brined olives treated with 1-MCP, SS-C: control Spanish-style olives, SS-1MCP: Spanish style olives treated with 1-MCP. Time 0 corresponds to the moment just after putting the fruits in brine fermentation. Non-overlapping error bars stand for significant differences among treatments according to ANOVA analysis and post hoc statistical LSD test using $p \leq 0.05$ as the cut-off level of significance.

Figure 3 shows the evolution of different parameters determining the fruits' quality. A typical texture and green colour (>hue angle) are desired at the end of the respective fermentation processes. The final firmness of the fermented fruit is greatly affected by the initial texture of raw material, with the primary cell wall and middle lamella structure and composition being the main factors shaping this parameter evolution [32]. In Figure 3, sensible differences were observed between SS and DB fruits. The application of NaOH during processing produced a considerable loss of texture but, by contrast, preserved the fresh appearance of fruits compared to directly brined olives [2]. Applying 1-MCP slightly increased pulp firmness in both SS and DB treatments compared to their respective controls (Figure 3A), whilst between 1-MCP treated and untreated fruits, differences in retention of green colour determined as hue angle was more evident in the case of DB-1MCP treatment. As is known, the skin colour of olives is greatly affected by the synthesis of anthocyanin during ripening, as also observed during postharvest handling. The anthocyanins are formed when the first pink spots appear in colour-turning olives, and their content increases as maturation progresses, with cyaniding-3-O-glucoside and cyaniding-3-O-rutinoside mainly responsible for the colour [33]. Yoruzmaz et al. [34] measured the evolution of anthocyanins during maturation, observing a considerable and rapid increase as the surface

colour became pink. Ramin [16] and Amini & Ramin [15] also showed that applying 1-MCP to green olives favoured the retention of olive firmness and flesh appearance during postharvest storage for up to 15 weeks. Kafkaletou and Tsantili [9] showed that using 1-MCP in harvested dark green Conservolea olives prevented the green loss of fruits. However, applying 1.5 µL/L of 1-MCP to green-harvested Conservolea fresh olives did not improve firmness during 10 days of postharvest storage [35].

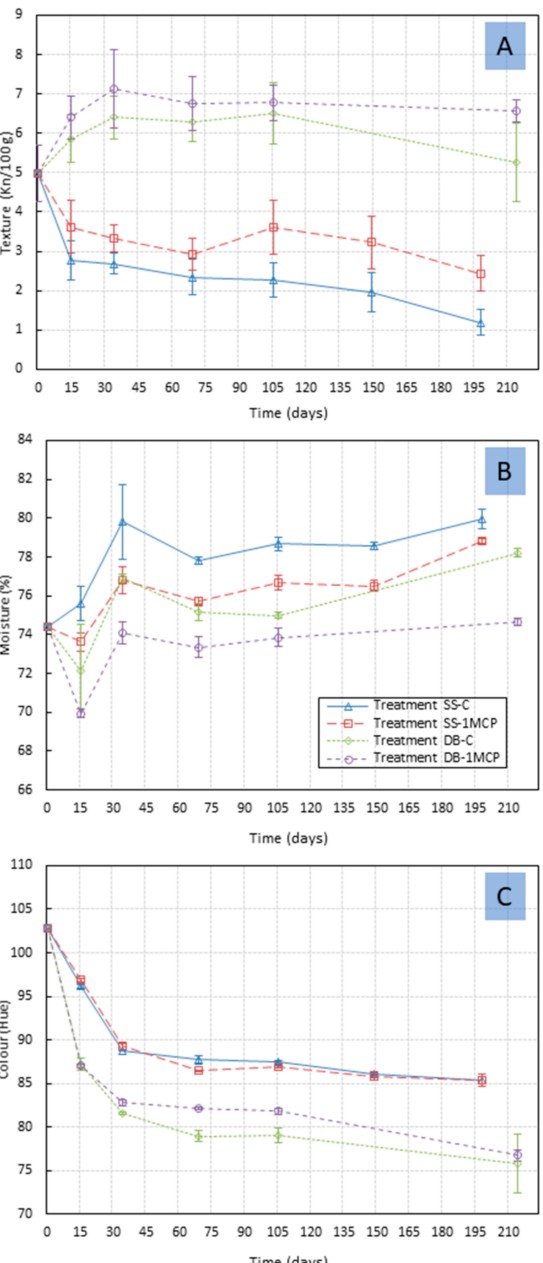

**Figure 3.** Evolution in fruits of instrumental texture (**A**), humidity (**B**), and instrumental colour measured as hue angle (**C**) during fermentation in the different treatments assayed in this work. DB-C: control directly brined olives, DB-1MCP: directly brined olives treated with 1-MCP, SS-C: control Spanish-style olives, SS-1MCP: Spanish style olives treated with 1-MCP. Time 0 corresponds to the moment just after putting the fruits in brine fermentation. Non-overlapping error bars stand for significant differences among treatments according to ANOVA analysis and post hoc statistical LSD test using $p \leq 0.05$ as the cut-off level of significance.

Because of the normal development of the fermentation processes, the total reducing sugars in brine were only determined at the end of the fermentation (176 days). The only sugar detected was mannitol, with a concentration of 0.17 (SD, 0.05), 0.15 (0.01), 2.41 (1.07), and 2.14 (0.15) g/L for SS, SS-1MCP, DB, and DB-1MCP treatments, respectively. Thus, the mannitol concentration was higher in DB than in SS elaborations but without significant effect due to the application of 1-MCP. Bautista-Gallego et al. [36] also reported that mannitol is not the sugar preferentially consumed by microorganisms during olive fermentation and packaging.

### 3.2. Microbiological Changes

The evolution of the LAB, yeasts and *Enterobacteriaceae* populations in brine during 176 days of fermentation was usual for both types of olive processing (Figure 4), without significant differences between 1-MCP treated and untreated fruits. In fact, the maximum LAB population was obtained in brine on the 12th day of fermentation in SS elaboration (lye treated olives) at approximately 8.8 $\log_{10}$ CFU/mL, whilst in DB fermentations, the maximum population was obtained on the 34th day of fermentation at 7.7 $\log_{10}$ CFU/mL (Figure 4A). On the other hand, DB fermentations favoured yeast growth, with the maximum population (approximately 7.0 $\log_{10}$ CFU/mL) being obtained after the 16th day of fermentation, in contrast with SS fermentation, with only 6 $\log_{10}$ CFU/mL on the 22nd day of fermentation (Figure 4B). In directly brined processes, hydrolysis of phenolic compounds is achieved more slowly than in lye-treated olives because of the absence of NaOH hydrolytic action [3]. Moreover, many of these phenolic compounds are potent antibacterial compounds that hinder the growth of LAB species during olive fermentation [37]. Therefore, the growth of LAB species in the directly brined process is more limited than in lye-treated olives. By contrast, yeasts are more resistant to phenolic compounds, so their growth was more outstanding in directly brined olives [38]. *Enterobacteriaceae* were only detected in DB fermentations, reaching populations between 5–6 $\log_{10}$ CFU/mL during 6–47 days of fermentation (Figure 4C). Throughout this period, the pH of fermentation was around 5.0 units (Figure 2A), which could favour their survival [2].

Finally, the number of microorganisms forming biofilms was also determined at the end of fermentation. The LAB counts were 6.71 (SD, 0.37), 6.60 (0.13), 5.35 (1.74), and 6.51 (0.45) $\log_{10}$ CFU/g for SS, SS-1MCP, DB, and DB-1MCP treatments, respectively. In the case of yeasts counts, they were 4.35 (0.12), 2.74 (0.22), 5.74 (1.35), and 4.71 (0.15) $\log_{10}$ CFU/g for SS, SS-1MCP, DB, and DB-1MCP treatments, respectively. *Enterobacteriaceae* were only detected at the end of fermentation in the biofilms of DB control treatment at 2.29 (3.24) $\log_{10}$ CFU/g. Studying microorganisms associated with olive epidermis is more complex than in brines because detachment of cells from mature biofilms may be incomplete. Therefore, microbial counts would be underestimated. In the present work, microbial counts obtained in olive biofilms were similar to those obtained in previous studies [39], which obtained more than 6 million LAB and 3500 UFC of yeasts per gram of olives on the 90th day of fermentation. Results also showed that 1-MCP did not affect the microbiological course of fermentation, and there was no indication of 1-MCP toxicity, as is the case in citrus [40].

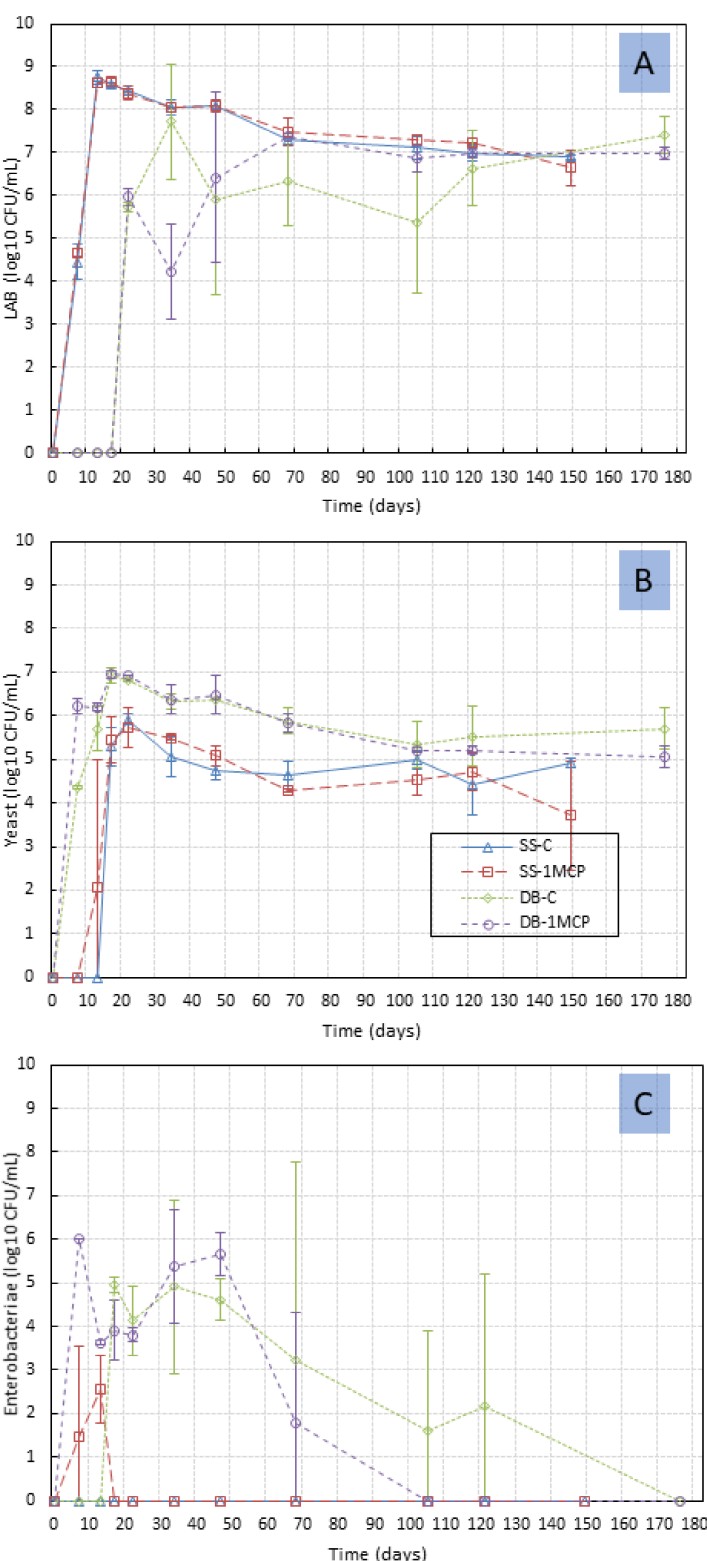

**Figure 4.** Evolution in brines of LAB (**A**), yeasts (**B**), and *Enterobacteriaceae* (**C**) populations during fermentation in the different treatments assayed in this work. DB-C: control directly brined olives, DB-1MCP: directly brined olives treated with 1-MCP, SS-C: control Spanish-style olives, SS-1MCP: Spanish style olives treated with 1-MCP. Time 0 corresponds to the moment just after putting the fruits in brine fermentation. Non-overlapping error bars stand for significant differences among treatments according to ANOVA analysis and post hoc statistical LSD test using $p \leq 0.05$ as the cut-off level of significance.

### 3.3. Metagenomic Analysis

A meta-taxonomics analysis was also carried out to determine the bacterial and fungal taxonomy in the biofilms at the end of the fermentation period. In the 16S rRNA amplicon study (bacteria), 1,175,752 raw sequences were generated for the eight samples analysed, with a mean of 146,969 reads per sample. After quality and chimera filtering, 35.81% of the reads were retained. Filtered 16S sequences were assigned to ten different bacterial ASVs. They represented 70.02%% of the total sample composition, while the rest were chloroplast (29.78%), mitochondrial (0.12%), and unclassified (0.08%) reads. The eight ITS (fungi) amplicon samples generated 1,253,961 raw reads with a mean of 156,745 reads per sample. After quality and chimera filtering, 20.17% of them were retained. Filtered ITS sequences were assigned to ten different fungi ASVs. They represented 84.60% of the total sample composition. Unclassified ASVs, with no correspondence in the UNITE database, accounted for 15.39% of the total composition. Large differences were found in the number of chloroplasts between SS and DB olives, reaching values higher than 50% of chloroplast sequences in the first case, compared to 8% in the second, which translated into a lower percentage of assigned ASVs sequences in the SS olives. Moreover, 25.92% of fungal sequences of SS olives could not be assigned to ASVs compared to 2.24% of DB olives. As expected, SS fermented fruits can present a lower texture than the natural olives due to the NaOH treatment, which can lead to a more outstanding breakage of the cellular structure and, with it, greater extraction of chloroplast DNA. No differences between olives treated with 1-MCP and controls were observed.

As an overall comparison, Table 1 shows the relative abundance of bacterial and fungal populations at the genus levels, grouping all samples as a function of the type of elaboration (SS or DB) and treatment (control or 1-MCP). Bacterial biodiversity was lower than fungal. Among the bacterial populations, *Lactiplantibacillus* was the predominant genus in all samples, ranging from 72.5 (DB-1MCP) to 91.54% (SS-C), with a major presence in the fruits elaborated as SS. *Lactiplantibacillus pentosus* and *Lactiplantibacillus plantarum* were the majority LAB species isolated from table olive fermentations and packagings [41,42]. In this case, the genus was also inoculated at the onset of the fermentation process. Another minority bacterial genus also present in all samples was *Pediococcus*, ranging from 2.4% (SS-C) to 10.1% (DB-1MCP), but in this case, its presence was higher in green DB fruits. Benítez-Cabello et al. [42] also reported *Pediococcus* as a LAB genus widely distributed in diverse table olive presentations. No statistical differences were observed between 1-MCP treated and untreated fruits for both bacterial genera. The presence of genera *Erwinia*, *Celerinatantimonas*, *Enterococcus*, and *Halomonas* was sporadic (<0.2%) and only associated with one specific sample. By contrast, *Enterobacter* was detected in a higher proportion in DB-1MCP samples (15.6%) compared to the same control treatment (DB-C, 0.04%) (Table 1).

Regarding fungi taxonomy, six genera (*Saccharomyces*, *Wickerhamomyces*, *Zygoascus*, *Candida*, *Aureobasidium*, and *Cladosporium*) were detected in all samples, but their frequencies were significantly different between SS and DB olives. *Saccharomyces* was the predominant yeast in DB treatments (>97%), whilst its presence in SS fruits was lower (0.2–5.7%). On the other hand, *Zygoascus* and *Wickerhamomyces* (especially) were the yeast genera predominant in SS fermentations. The presence of both microorganisms was very scarce in DB olives (0.1–0.8%). Applying 1-MCP to the SS fruits doubled the presence of *Wickerhamomyces* genus (74.46%) whilst reducing *Zygoascus* (mainly) and *Saccharomyces* (Table 1). *Saccharomyces*, *Wickerhamomyces*, and *Zygoascus* are yeast species frequently isolated from diverse table olive processing types [38,43].

**Table 1.** Relative abundance (%) in olive biofilms of bacterial and fungal genera at the end of the fermentation process (176 days).

| Bacterial genera | SS-C | SS-1MCP | DB-C | DB-1MCP |
|---|---|---|---|---|
| *Lactiplantibacillus* | 91.54 (4.06) | 94.52 (5.32) | 87.44 (7.01) | 72.46 (9.32) |
| *Enterobacter* | 0.00 (0.00) | 0.00 (0.00) | 0.04 (0.06) | 15.55 (21.64) |
| *Paraliobacillus* | 5.79 (5.91) | 2.66 (3.19) | 0.00 (0.00) | 0.00 (0.00) |
| *Mangrovibacter* | 0.00 (0.00) | 0.00 (0.00) | 2.78 (0.80) | 1.88 (1.86) |
| *Pantoea* | 0.00 (0.00) | 0.00 (0.00) | 0.07 (0.10) | 0.00 (0.00) |
| *Enterococcus* | 0.20 (0.29) | 0.00 (0.00) | 0.00 (0.00) | 0.00 (0.00) |
| *Halomonas* | 0.00 (0.00) | 0.04 (0.06) | 0.00 (0.00) | 0.00 (0.00) |
| *Celerinatantimonas* | 0.04 (0.05) | 0.00 (0.00) | 0.00 (0.00) | 0.00 (0.00) |
| *Erwinia* | 0.00 (0.00) | 0.00 (0.00) | 0.00 (0.00) | 0.01 (0.01) |
| *Pediococcus* | 2.41 (1.61) | 2.75 (2.19) | 9.65 (6.37) | 10.08 (10.58) |

| Fungal genera | SS-C | SS-1MCP | DB-C | DB-1MCP |
|---|---|---|---|---|
| *Saccharomyces* | 5.71 (3.58) | 0.25 (0.35) | 98.78 (1.02) | 97.82 (0.40) |
| *Cystobasidium* | 0.00 (0.00) | 0.17 (0.23) | 0.00 (0.00) | 0.00 (0.00) |
| *Meyerozyma* | 0.00 (0.00) | 0.00 (0.00) | 0.02 (0.02) | 0.00 (0.00) |
| *Holtermanniella* | 0.00 (0.00) | 0.00 (0.00) | 0.02 (0.02) | 0.00 (0.00) |
| *Dipodascus* | 0.00 (0.00) | 0.82 (1.15) | 0.00 (0.00) | 0.00 (0.00) |
| *Itersonilia* | 0.59 (0.84) | 0.00 (0.00) | 0.02 (0.02) | 0.00 (0.00) |
| *Rhodotorula* | 0.45 (0.64) | 0.14 (0.19) | 0.01 (0.01) | 0.00 (0.00) |
| *Sigarispora* | 0.00 (0.00) | 0.00 (0.00) | 0.01 (0.01) | 0.00 (0.00) |
| *Pichia* | 0.33 (0.46) | 0.00 (0.00) | 0.00 (0.00) | 0.03 (0.01) |
| *Sistotrema* | 0.19 (0.26) | 0.00 (0.00) | 0.00 (0.00) | 0.00 (0.00) |
| *Cercospora* | 0.14 (0.20) | 0.00 (0.00) | 0.00 (0.00) | 0.00 (0.00) |
| *Hanseniaspora* | 0.00 (0.00) | 0.05 (0.06) | 0.00 (0.00) | 0.00 (0.00) |
| *Pleurophoma* | 0.00 (0.00) | 0.00 (0.00) | 0.00 (0.00) | 0.01 (0.00) |
| *Wickerhamomyces* | 36.95 (25.51) | 74.46 (29.91) | 0.77 (0.82) | 0.22 (0.12) |
| *Zygoascus* | 36.31 (43.35) | 17.73 (24.57) | 0.07 (0.10) | 0.29 (0.01) |
| *Candida* | 3.08 (1.95) | 0.40 (0.56) | 0.16 (0.01) | 0.92 (0.05) |
| *Aureobasidium* | 3.13 (0.70) | 2.04 (1.51) | 0.06 (0.04) | 0.20 (0.15) |
| *Cladosporium* | 2.81 (2.93) | 1.01 (0.99) | 0.02 (0.01) | 0.12 (0.07) |
| *Debaryomyces* | 0.00 (0.00) | 0.00 (0.00) | 0.00 (0.00) | 0.10 (0.14) |
| *Naganishia* | 2.62 (3.70) | 0.00 (0.00) | 0.01 (0.01) | 0.00 (0.00) |
| *Nakazawaea* | 0.00 (0.00) | 0.00 (0.00) | 0.00 (0.00) | 0.12 (0.16) |
| *Uncobasidium* | 2.30 (3.26) | 0.00 (0.00) | 0.00 (0.00) | 0.00 (0.00) |
| *Saccharomycopsis* | 0.00 (0.00) | 0.00 (0.00) | 0.00 (0.00) | 0.06 (0.08) |
| *Vishniacozyma* | 1.81 (2.56) | 0.00 (0.00) | 0.00 (0.00) | 0.02 (0.02) |
| *Sporobolomyces* | 0.47 (0.67) | 1.12 (1.59) | 0.00 (0.00) | 0.01 (0.01) |
| *Botryosphaeria* | 1.03 (1.49) | 0.00 (0.00) | 0.00 (0.00) | 0.00 (0.00) |
| *Dekkera* | 0.00 (0.00) | 0.35 (0.48) | 0.00 (0.00) | 0.00 (0.00) |
| *Schwanniomyces* | 0.00 (0.00) | 1.21 (1.70) | 0.03 (0.04) | 0.02 (0.02) |
| *Priceomyces* | 0.52 (0.72) | 0.23 (0.33) | 0.00 (0.00) | 0.04 (0.06) |
| *Malassezia* | 1.50 (1.42) | 0.00 (0.00) | 0.00 (0.00) | 0.00 (0.00) |

Note: Only the most abundant sequences (>0.01%) with presence in at least one sample are shown. SS-C: Control Spanish Style olives, SS-1MCP: Spanish style olives treated previously with 1-MCP, DB-C: control directly brined olives, DB-1MCP: directly brined olives previously treated with 1-MCP. Standard deviation in parentheses (*n* = 2).

*3.4. Sensory Evaluation*

With regards to the sensory evaluation (Table 2), there were no significant differences ($p \geq 0.05$) among treatments for the acidic, salty, and defective attributes. However, there were significant differences ($p \leq 0.05$) in hardness, bitterness, browning, and overall acceptability. Hardness, bitterness, and browning scores were higher in DB olives compared with SS packaged fruits. However, the overall acceptability score was higher for the SS treatments (>5.4) than in the case of DB olives (<4.5). Applying 1-MCP increased scores for the hardness attributes and overall acceptability in both SS and DB olives (Table 2), notably in the case of SS-1MCP fruits.

**Table 2.** Scores assigned by the 10 expert panellists to the sensory evaluation of the various treatments packaged in this work.

| | Treatment | | | |
|---|---|---|---|---|
| Attribute | SS-C | SS-1MCP | DB-C | DB-1MCP |
| Hardness | 2.71 (0.49) a | 5.62 (0.99) b | 6.91 (0.58) c | 7.15 (0.57) c |
| Acidic | 5.03 (0.62) a | 5.76 (0.70) a | 5.22 (0.52) a | 4.83 (0.48) a |
| Salty | 5.07 (0.49) a | 5.14 (0.41) a | 5.01 (0.25) a | 4.83 (0.44) a |
| Bitterness | 2.60 (1.13) a | 3.02 (0.81) a | 6.52 (0.62) b | 6.13 (0.83) b |
| Browning | 0.35 (0.31) a | 0.32 (0.30) a | 4.90 (0.46) b | 5.33 (0.55) b |
| Flavour/aroma defects | 0.70 (0.35) a | 0.96 (0.43) a | 0.97 (0.32) a | 0.90 (0.29) a |
| Overall acceptability | 5.45 (0.96) a | 6.86 (1.52) b | 4.36 (1.45) c | 4.59 (0.99) c |

Note: SS-C: Control Spanish Style olives, SS-1MCP: Spanish style olives treated previously with 1-MCP, DB-C: control directly brined olives, and DB-1MCP: directly brined olives previously treated with 1-MCP. Standard deviation in parentheses ($n = 10$). Values followed by different letters within the same row are statistically different ($p \leq 0.05$) according to the LSD post hoc comparison test.

### 3.5. Multiple Factor Analysis

Multiple Factor Analysis (MFA) is devoted to data tables in which a set of individuals is described by several groups of variables [27]. In this case, the groups were the physicochemical characteristics of the brines at the end of fermentation (pH, titratable acidity, combined acidity, and NaCl content), the features of the fruits at the end of fermentation (texture, CIE LAB parameters, and moisture), the proportions of the diverse species of bacteria and fungi found after the metagenomic analysis, and the attributes of the organoleptic evaluation (browning, bitterness, hardness, acidic, salty, flavour/aroma defects, and overall acceptability). Since the multivariate analysis does not allow row replicate names, the study, in this case, was performed on the average values per treatment of the different variables. An MFA provides representations of the individual and variables which can be interpreted similarly to Principal Components Analysis. As shown in Figure 5A, considering all the characteristics simultaneously, one should consider the treatments as grouped into three clusters associated with dissimilarities in the type of processing. Within each style, the application of 1-MCP only had a significant effect on SS in which the product application led to a different fungal population and a product with the highest acceptability; however, 1-MCP hardly affected the directly brined (natural) process and final products. The factor map (Figure 5A) and the correlation circle of the variables (Figure 5B) were plotted separately, to improve visualisation, but their interpretation should be based on both since they are two complementary aspects of the analysis. The variables (Figure 5B) appearing on the same side (high value) as one treatment (Figure 5A) are associated with it. By contrast, the variables on the opposite side are not linked. Therefore, DB-C and DB-1MCP (cluster 1, on the left of Figure 5A) showed very similar characteristics among them, which could be associated with browning, bitterness, high instrumental and sensory hardness, high values of parameter a* (colour) and the presence of *Saccharomyces* and *Mangrovibacter* (Figure 5B). On the other side, SS-C and SS-1MCP were quite different from the DB olives but differed strongly between them: the 1-MCP treatment strongly influenced the fermentation leading to two distinct clusters (2 and 3) because of the brine and fruit characteristics. SS-C was related to the presence of *Paraliobacillus* and several fungal genera (*Rhodotorula, Cladosporium, Zygoascus,* and *Priceomyces*). However, those SS olives treated with 1MCP (SS-1MCP) were mainly characterised by the growth of *Sporobolomyces* and *Wickerhamomyces* and were the most appreciated product since their fruits received the highest overall acceptability. However, both SS processes share most of the properties related to colour (b*, c*, h and L), combined acidity and NaCl contents, and the growth of *Aureobasidium*. Thus, the MFA is an elegant method for summarising the influence of 1-MCP treatment and the processing

style on the fermentation and characteristics of the final products. The correlation circle also establishes the relationships among variables. Brown, bitter and hard olives have high values of a* parameter, for texture; moreover, *Saccharomyces* and *Mangrovibacter* are abundant in their fermentations. By contrast, olives with high values of L, b*, c*, and *hue* have marked acidity and are associated with *Aureobasidium, Zigoascus* or *Paraliobacillus*; in addition, they enjoy good acceptability.

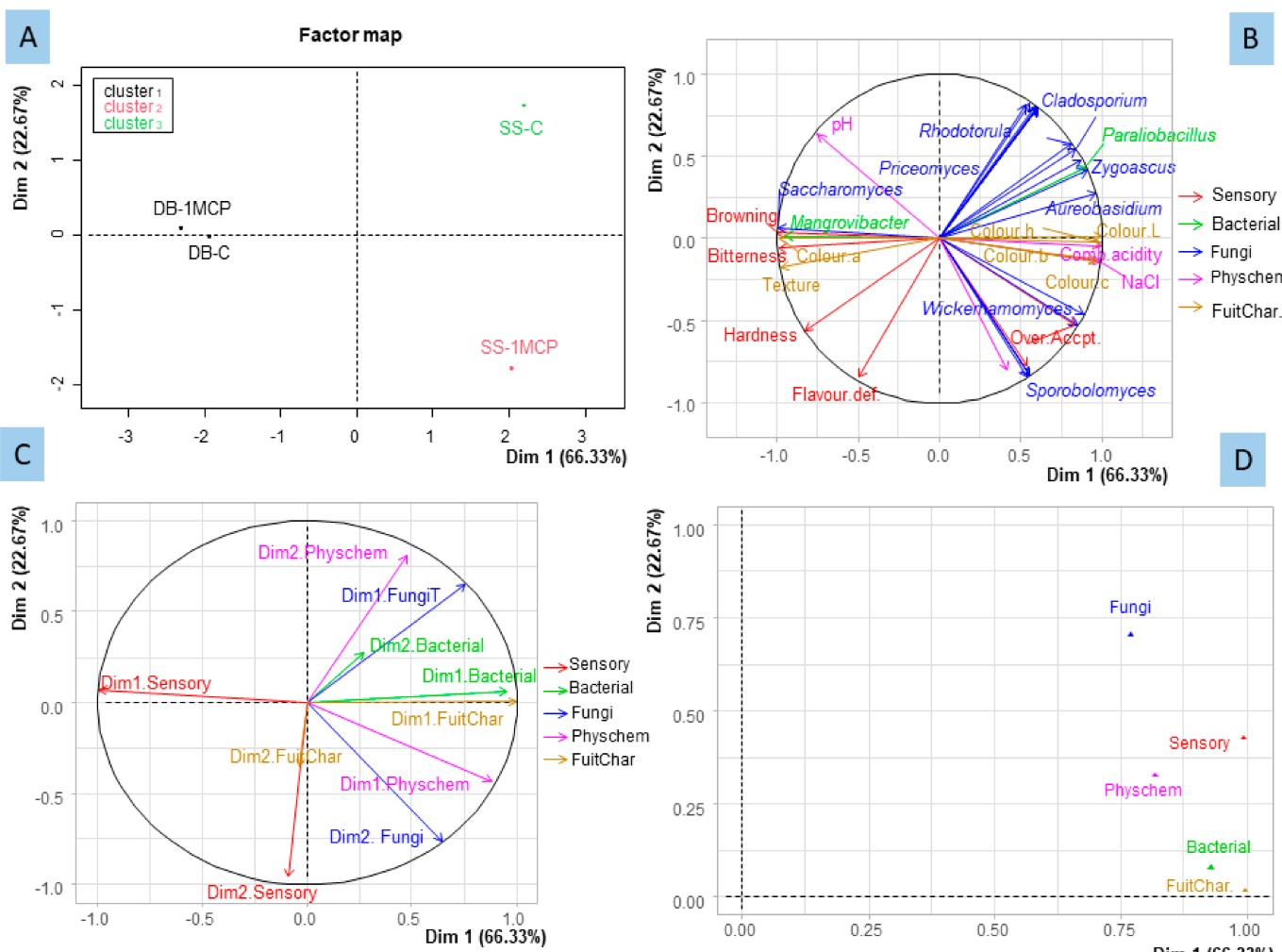

**Figure 5.** Factor map of treatments on the plane of the first two overall dimensions, showing simultaneously their automatic clustering (**A**), correlation circle of different sensory, microbial, physic-ochemical, and fruit characteristics (**B**), comparison of the alignment of overall and each group's first two dimensions (**C**), and projection of the groups on the plane of the first two overall dimensions (**D**). The analysis was based on the average values per treatment of the different variables.

The MFA may also provide a way of comparing the impact of each group on the overall trend by projecting the two first dimensions of each group on that of the general PCA (Figure 5C). The trends of Dim 1 of bacterial and fruit characteristics are in the same direction as general PCA Dim 1, indicating a good agreement. In addition, Dim 1 for physicochemical characteristics of the brine and fungi also show a marked degree of alignment. On the other hand, the sensory attribute is the only group that shows an opposed (inverse) trend, but a high agreement in absolute terms. This agreement between the overall trend (dominant dimension) and those of the groups (Figure 5C) is also reflected by the position of the different groups on the overall map (Figure 5D), which is, in fact, their projections on the general dimensions. This map of groups shows that all of them are closely

linked to Dim1, indicating that this is an important direction of inertia (or variance) for all of them. The fruit characteristics and bacterial groups present a high agreement with Dim 1, without a practical relationship with Dim 2. Physicochemical and sensory characteristics also have most of their information aligned with Dim 1 but increase its association with Dim 2. Finally, the fungi group is strongly linked to Dim 1 but also has a marked projection on Dim 2, as deduced from its positions on overall dimensions (Figure 5D).

## 4. Conclusions

Applying 1-MCP to Manzanilla fruits processed as SS and DB olives reduced the colour-turning of olives, which is important in case of processing delay, and did not affect the microbiological course of fermentation. Indeed, after fermentation, physicochemical parameters such as pH and titratable acidity in brine and the colour and texture of fruits improved. The 1-MCP treated olives also had better overall acceptability in the sensory evaluation of packaged fruits, especially in the Spanish-style processing. This work opens the possibility of using 1-MCP as a pretreatment in the elaboration of table olives. However, further studies should be conducted to optimize its concentration and interaction with other variables which govern olive fermentation (temperature, salt, pH, or acid addition).

**Author Contributions:** Conceptualisation, A.G.-F.; methodology, E.L.-G., V.M.-A. and A.B.-C.; analysis and data curation, A.G.-F., F.N.A.-L. and A.B.-C.; writing and review, F.N.A.-L., A.B.-C., F.R.-G. and A.G.-F.; funding, F.N.A.-L. All authors have read and agreed to the published version of the manuscript.

**Funding:** This research was funded by MCIU/AEI/FEDER, UE, grant number TOBE project RTI-2018-100883-B-I00.

**Institutional Review Board Statement:** Not applicable.

**Informed Consent Statement:** Not applicable.

**Data Availability Statement:** Not applicable.

**Acknowledgments:** We acknowledge the support of the publication fee by the CSIC Open Access Publication Support Initiative through its Unit of Information Resources for Research (URICI). Author E.L.-G. thanks the Spanish Ministry of Science and Innovation for his FPI contract (PRE2019-087812). Author A.B.-C. thanks the Junta de Andalucía for his postdoctoral contract (PAIDI2020-00162). Author A.G.-F. thanks the CSIC for his "Ad honorem" appointment. We also thank Eve Dupille, R&D Manager of Agrofresh, for providing the 1-MCP compound for the experiments, Jolca S.A., and especially Juan Carlos Roldán and Rosa Torres for providing the fruits.

**Conflicts of Interest:** The authors declare no conflict of interest.

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
