# Peer review of "Influence of 1-Methylcyclopropene (1-MCP) on the Processing and Microbial Communities of Spanish-Style and Directly Brined Green Table Olive Fermentations"

_fermentation, doi:10.3390/fermentation8090441_

Round 1

Reviewer 1 Report

The manuscript „fermentation-1847984” presents the results of a study regarding the influence of 1-methylcyclopropene (1-MCP) on the processing of olive fruits using alkaline and natural brine treatments. 1-MCP was chosen because it is an effective inhibitor of ethylene action in fruit. 1-MCP proved to delay the ripening or the senescence process and extend the shelf life of several fruits.

The authors treated half of the freshly harvested green olives with 2.85 μL/L 1-MCP and kept the rest as control. The changes in colour and damaged fruits during one and 21 days of storage were determined on one kg of untreated and one kg of 1-MCP-treated olives. Afterwards, the rest of the olives were subjected to alkaline treatment (Spanish-style olives) or non-acidic brine (5 % NaCl solution). Olives treated with alkali were washed to remove the alkali in excess and then brined in an acidic 11 % NaCl solution. All the samples were run in duplicate and inoculated with a commercial starter containing three strains of Lactiplantubacilluss pentosus, and another starter containing the yeasts Wickerhanomyces anomalus and Saccharomyces cerevisiae. Samples were taken at different fermentation moments to monitor the physicochemical properties: pH, NaCl concentration, titratable and combined acidity, firmness, surface colour, and moisture content. Microbial analysis was focused on counting the Enterobacteriaceae, yeasts, and lactic acid bacteria. Sensory evaluation was applied after 176 days (22 days) of fermentation, scoring acidic, salty, bitterness, hardness, browning, appreciation of defects, and overall acceptability. Metagenomic analysis was performed at the end of fermentation to determine the structure of the bacterial and fungi populations. The treatments with 1-MCP changed the proportions of microorganisms, e.g., the presence of Wickerhamomyces in Spanish-style olives doubled. Also, the presence of Zygoascus was reduced, allowing the growth of Enterobacter in natural brined olive fruits. 1-MCP treated fruits showed lower pH levels, higher titratable acidity and firmness, and better colour than untreated fruits. The study concluded that 1-MCP postharvest treatment of olives reduced the maturation process and improved the quality and sensory attributes of table olives.

The manuscript is well organised and written, and the results support the conclusions. However, a few minor errors were detected and presented below.

Small observations and suggestions

L246 Split the sentence: replace the comma after „(Figure 1A, 1C) with a dot and „but” with „Still,”.

L256, 535 Correct „physic-chemical”. Suggestion: use „physicochemical” as in L80, 109, 155, 213, 241, 481, 508, and 517.

L476 „Standard deviation in parentheses (n=10).” appears to be a sentence fragment. Consider rewriting it as a complete sentence.

L533 Insert the preposition „of” between „colour” and „olives”.

Author Response

We sincerely thank reviewer 1 for his/her constructive comments, which in our opinion, have improved the quality and clarity of the revised manuscript. Please, find below a detailed response point by point to their queries.

Q1. L246 Split the sentence: replace the comma after „ (Figure 1A, 1C) with a dot and „but” with „Still,”.

A1. The sentence has been corrected.

Q2. L256, 535 Correct „physicochemical”. Suggestion: use „physicochemical” as in L80, 109, 155, 213, 241, 481, 508, and 517.

A2. The change was introduced according to the reviewer’s suggestion.

Q3. L476 „Standard deviation in parentheses (n=10).” appears to be a sentence fragment. Consider rewriting it as a complete sentence.

A3. Thank to reviewer for his/her suggestion. The sentence was corrected.

Q4. L533 Insert the preposition „of” between „colour” and „olives”.

A4. The preposition was inserted.

Reviewer 2 Report

The effect of 1-MCP and SS or DB treatments on the physicochemical, microbial, and sensory changes of green table olives were observed in this study. It can provide some useful information for the olive processing industrially. The manuscript is well organized and written. However, there are still some questions need to notice.

1.      The materials and methods: in the section 2.1 it was descripted eighty kg olives were used for study, and it was divided into four groups, namely SS, DB,and them with 1-MCP. In line 108, it says the experiment were duplicated (n=8). It is not clearly about the repetition experiments? Please clearly present.

2.      In the materials and methods section, it is not clear what is the 0 hour? As it has written, some olives were treated with 1-MCP for 20h, so please clearly describe the 0 hour, and how about the next 24 hours, and so on.

3.      As it has described the results were ANOVA, so please provide the results in the figures 1-4.

4.      The results and discussion: the authors found the difference between pH, acidity of treatments samples. But the discussion about the relationship between them, and between combined acidity and titratable acidity. And the significant difference show be explained.

5.      Section 2.2: As we have seen that the change of olives color was different clearly, but the samples will affect the color result. So the authors should clearly describe how to sample the olives of different treatments.

6.      In Figure 5, the Genus of bacteria and fungi is not clearly showed.

7.      In Figure 6, as concerned figure 6A every treatment only showed one point for the sign? As we all know that about the result there is usually above 2 sample signs for each treatment. The relationship of Figure 6B,6C and 6D need to discussion. Or maybe it is better to combine some digit result to discussion the results.

Author Response

We sincerely thank reviewer 2 for his/her constructive comments, which in our opinion, have improved the quality and clarity of the revised manuscript. Please, find below a detailed response point by point to their queries.

Q1. The materials and methods: in the section 2.1 it was descripted eighty kg olives were used for study, and it was divided into four groups, namely SS, DB,and them with 1-MCP. In line 108, it says the experiment were duplicated (n=8). It is not clearly about the repetition experiments? Please clearly present.

A1. Thank to reviewer for his/her suggestion. The sentence was clarified.

Q2.   In the materials and methods section, it is not clear what is the 0 hour? As it has written, some olives were treated with 1-MCP for 20h, so please clearly describe the 0 hour, and how about the next 24 hours, and so on.

A2. For Figures 2-4, the time 0 corresponds to the moment when putting the fruits in brines and beginning of fermentation, except for the evolution of turning colour olives and damaged fruits after harvesting (Figure 1), where time 0 corresponds to the moment just after picking. This point was clarified in the revised version of manuscript.

Q3. As it has described the results were ANOVA, so please provide the results in the figures 1-4.

A3.  According to reviewer’s suggestion, information of ANOVA analysis was introduced in Figure 1-4 legends.

Q4.   The results and discussion: the authors found the difference between pH, acidity of treatments samples. But the discussion about the relationship between them, and between combined acidity and titratable acidity. And the significant difference show be explained.

A4. This point was discussed in the revised version of manuscript.

Q5.   Section 2.2: As we have seen that the change of olives color was different clearly, but the samples will affect the color result. So the authors should clearly describe how to sample the olives of different treatments.

A5. The sentence has been clarified.

Q6.   In Figure 5, the Genus of bacteria and fungi is not clearly showed.

A6. Thank you to reviewer for his/her constructive comment. However, Figure 5 has been removed from the final version of manuscript because we have noticed that all information was already provided in table 1 and it was redundant.

Q7.   In Figure 6, as concerned figure 6A every treatment only showed one point for the sign? As we all know that about the result there is usually above 2 sample signs for each treatment. The relationship of Figure 6B, 6C and 6D need to discussion. Or maybe it is better to combine some digit result to discussion the results.

A7. The multivariate analysis does not allow the repetition of row names. In this case, MFA was performed on the average values of replicates to prevent such a problem. A paragraph explaining this circumstance was introduced in the text (L493-495 of the revised version). Besides, a short sentence indicating that the analysis was performed on average values per treatment of the different variables was introduced at the end of the legend of Figure 6 (revised version).  Besides, Figures 6B, C, and C have been commented on more extensively, emphasising their relationships. On the other hand, the main objective of the multivariate analysis is the projection of a set of variables in a reduced dimension (2 or 3 Dim) so that the main trends can be visualised. It could be possible introducing the correlation of the variables or dimensions of the groups with the new axes, but in some way, the information would be redundant since it is also observed from the figure.

Round 2

Reviewer 2 Report

it has been revised properly, it has arrived at the acceptance level.